# Humate Combined with Film-Mulched Ridge-Furrow Tillage Improved Carbon Sequestration in Arid Fluvo-Aquic Soil

**Fengke Yang** [1,2,*], **Baolin He** [1,2] and **Guoping Zhang** [1]

1   Dryland Farming Institute, Gansu Academy of Agricultural Sciences, Lanzhou 730070, China;
    blhe@163.com (B.H.); zhangguoping79@126.com (G.Z.)
2   Key Laboratory of High Water Utilization on Dryland of Gansu Province, Lanzhou 730070, China
*   Correspondence: yang_fk@163.com

**Abstract:** Commercial humic fertilizers (humates) can be used to improve carbon sequestration. In this study, a 3-year field trial (2016–2018) employed four treatments to investigate the mechanism by which humate increases carbon sequestration in fluvo-aquic soils: (1) blank: bare soil with no agricultural treatment; (2) control: standard film mulch (FM) ridge-furrow tillage (which acts as CK); (3) humate: FM tillage plus humate; and (4) straw: FM tillage plus straw. The three treatments strongly affected the soil carbon sequestration, with the humate and straw treatments more significant than the blank treatment. Moreover, the $\geq$2.0-mm macroaggregate fraction, >1-mm soil aggregate-associated carbon, weight mean diameter (MWD) and geometric mean diameter (GMD), and microbial biomass nitrogen (MBN) values for the straw and humate treatments were all significantly increased ($p < 0.05$), increasing the carbon sequestration by 1.9 and 0.9 Mg C ha$^{-1}$y$^{-1}$ compared to the control, respectively. Carbon sequestration was significantly associated with >1.0-mm aggregate-associated C, MWD, GMD, MBN, and organic C input. Humate and straw synergistically regulate the soil and microbial processes and greatly increase the straw C return to soil while efficiently increasing the macroaggregate fraction and stability, macroaggregate-associated carbon, and physical protection of aggregates, thereby increasing the carbon sequestration. Therefore, humate may be a novel economical alternative to straw to efficiently increase the carbon sequestration in dry fluvo-aquic soils.

**Keywords:** humate additive; carbon sequestration; macroaggregates; microaggregate-associated carbon; physical protection; full film-mulched ridge-furrow tillage

## 1. Introduction

Carbon sequestration is essential for the soil quality and crop productivity [1–3] and is determined by the input and output balance of the organic matter (e.g., crop residues, farmyard manure, organic manure, etc.) in agricultural soils [4–7]. Maintaining adequate carbon levels is the primary concern of long-term agricultural soil management for sustainable soil quality. Therefore, understanding the mechanism by which organic inputs improve soil carbon sequestration is of great importance, for both maintaining the soil quality and finding more economical and efficient additives to further increase the soil carbon content in arable land.

Straw is the most efficient soil improver among organic matter and is successfully used worldwide to maintain the soil carbon levels [8–10]. Many studies have shown that straw retention determines the soil carbon storage and stability [11] and controls the soil carbon dynamics [3,12–14] and, thus, greatly increases the soil organic carbon (SOC) sequestration, which mainly helps to enhance the formation of stable aggregates [15–17] and the physical protection of aggregates [18,19]. However, the positive effects are limited by the availability of straw, the type of straw retention, and the hydrothermal conditions of the soil [20,21]. Therefore, there is a need to find a novel substitute and to systematically investigate the mechanism by which it affects soil carbon storage.

Given the problem of straw scarcity, the use of commercial humate (a type of organic humic fertilizer) derived from leonardite as an organic additive could potentially be a useful way to improve soil carbon sequestration [2,22–24]. Humate contains humic substances (HS), which are among the most important sources of organic carbon in agroecosystems [23,25]. Previous studies have shown that HS (mainly humic acid and fulvic acid) can act as a permanent binder to promote soil aggregate formation [26,27] by increasing the aggregate carbon content and stability, which increases the soil carbon sequestration [28]. However, the use of commercial humate to improve soil carbon sequestration has been less-studied.

Plastic film mulch ridge-furrow (FM) tillage, a commonly used rainwater-harvesting technology, has been shown to be effective in sequestering carbon in rainfed arable land on the Loess Plateau [15,29]. Due to the increased return of organic material (crop residues, debris, and roots) to the soil from the vigorous plant growth promoted by the altered hydrothermal conditions in the soil, the soil carbon content increases significantly [30–32]. FM tillage also has some shortcomings. Several studies have found that FM tillage is associated with increased soil organic matter degradation and organic carbon mineralization. This is due to the prolonged high water and temperature conditions in the soil throughout the crop's growing season, which results in organic carbon consumption, thereby reducing the soil carbon content [20,21]. Therefore, maintaining the soil carbon levels in soil with FM tillage is quite difficult [33]. To maintain the SOC balance, continuous efforts are required to put into place the proper management procedures. Thus, the combination of organic improvement (especially humate addition) with FM tillage was developed, and the putative mechanisms underlying the effect on soil carbon sequestration require further investigation.

In this study, we hypothesized that humate combined with FM tillage could improve soil carbon sequestration in the short term by promoting the formation of stable macroaggregates and aggregate-associated carbon levels and increasing the physical protection of aggregates in soil carbon. The goals of this study were to investigate: (1) the distribution and stability of the soil aggregate fraction and aggregate-associated carbon content and dynamics, (2) the possible mechanism by which the configuration increases the soil carbon sequestration, and (3) the possibility of using commercial humate substitute straw as an organic soil improvement.

## 2. Methods

### 2.1. Experimental Sites

Field tests were conducted at the Zhuanglang Testing Station of the Gansu Academy of Agricultural Sciences (106°05′28″ E, 35°10′30″, 1765 m above sea level), which was established in 1997 to focus on long-term agricultural ecology and environmental studies in the Loess Plateau, China. The study region has a semi-arid continental monsoon climate, with an average annual temperature of 7.9 °C and 510 mm of rainfall. During the study period (2016–2018), the average daily minimum, mean, and maximum air temperature were −4.6, 8.3, and 34.6 °C, with annual precipitation of 416.0, 469.0, and 659.2 mm, respectively. At the test site, a plow with a working depth of 20–25 cm is frequently employed. The site has fluvo-aquic soil, which is similar to Aridisols in the USDA soil taxonomy [34] and contains loess raw material [35]. The properties of the soil in the depth range of 0–60 cm are shown in Table 1.

**Table 1.** Key properties of the soil layers (0–60 cm in depth).

| Soil Layer (cm) | Soil Texture | pH | SOC (g kg$^{-1}$) | AN (mg kg$^{-1}$) | AP (mg kg$^{-1}$) | AK (mg kg$^{-1}$) | TN (g kg$^{-1}$) | BD (g cm$^{-3}$) | >0.25-mm Soil Aggregates by Dry Sieving (%) | Carbonate Content (g kg$^{-1}$) |
|---|---|---|---|---|---|---|---|---|---|---|
| 0–20 | Clay loam | 8.5 | 7.1 a | 24.3 a | 25.6 a | 167.6 a | 0.80 a | 1.42 a | 64.5 | |
| 20–40 | Sandy loam | 8.5 | 6.8 b | 14.4 b | 4.1 b | 114.0 b | 0.57 b | 1.47 b | 69.4 | 90–180 (CaCO$_3$) |
| 40–60 | Sandy loam | 8.5 | 4.0 c | 13.0 c | 1.5 c | 91.2 c | 0.45 c | 1.62 c | 60.3 | |

Note: SOC, soil organic carbon; TN, total nitrogen; AN, alkali-hydrolysable nitrogen; AP, available phosphorus; AK, available potassium; BD, bulk density. Different letters (a, b, c) in the same column indicate significantly different means according to the LSD test at $p < 0.05$ level.

## 2.2. Study Design

The study protocol was developed in 2016 to study the soil C dynamics in the FM tillage system, i.e., narrow ridges (0.15 m high × 0.40 m wide) alternating with wide ridges (0.10 m high × 0.70 m wide), with the whole surface of the soil covered with a plastic film, typical for the cultivation of maize on the Loess Plateau. Then, we performed a 3-year field trial with several pretreatments: (1) blank: bare soil, no agricultural treatment; (2) control: standard FM practice using a standard fertilizer dose of 150 kg ha$^{-1}$ N, 120 kg ha$^{-1}$ P$_2$O$_5$, and 75 kg ha$^{-1}$ K$_2$O (which served as CK); (3) humate: FM plus "HengBo "granulated humate, i.e., commercial organic humus fertilizer of 1500 kg ha$^{-1}$ (organic matter/dry reading ≥45.0–55.0%, free humic acid/dry reading ≥20.0%) manufactured by HengBo Biotechnology Center, Lingshi County, Shanxi, China; and (4) straw: FM with straw retention, i.e., 7500 kg ha$^{-1}$ air-dried maize straw, chopped into 2 to 3-cm pieces, evenly incorporated into the 20–25-cm topsoil layer before the ridge-furrow and film mulching treatments. The 4 treatments were applied to a total of 12 plots in triplicate, with a plot size of 40 m$^2$ (5 × 8 m) separated by a 0.6-m walkway. Urea (46% N), superphosphate (12% P$_2$O$_5$), and K$_2$SO$_4$ (52% K$_2$O) were used as chemical fertilizers. We applied composted pig manure for several months as an organic fertilizer using the usual local application rates (N: 1.5–2.0 g kg$^{-1}$, P$_2$O$_5$: 0.80–2.50 g kg$^{-1}$, and K$_2$O: 231–291 mg kg$^{-1}$).

All chemical fertilizers were applied using the local protocols: one-third of the N and all of the P and K were used as the base fertilizer for each plot. The other two-thirds of the N fertilizer were applied on the corn shoots (55–60 days after sowing) in 10-cm rows between the plants using handheld implements. Organic fertilizer was used once a year before maize was sown.

## 2.3. Soil Sampling and Measurements

Soil samples were taken from each of the 4 treatments immediately after the corn was harvested in early October 2018 to measure the soil bulk density (BD), soil aggregate distribution, soil aggregate-associated organic carbon concentration, and microbial biomass nitrogen and carbon (MBN and MBC).

### 2.3.1. Soil Bulk Density (BD)

The soil bulk density (BD) at depths of 0–20, 20–40, and 40–60 cm in each plot was measured using the core method [36]; the volume of the stainless-steel ring used was 200 cm$^3$. Three core samples were randomly collected from each plot.

### 2.3.2. Soil Aggregate

The soil aggregate distribution was determined using the dry sieving method [13] in order to investigate the real situation at the site [11].

A total of 36 undisturbed soil samples were taken with a 5-cm diameter pipe auger at depths of 0–20, 20–40, and 40–60 cm. Stones, fibers, straw residue, and other visible organic parts were removed. Briefly, each sample (~200 g) was placed on a sieve (0.25 to 5 mm) and spun on an oscillator at 270 rpm for 2 min. The soil weight was measured and the ratio of aggregates to total mass was calculated. Since fluvo-aquic soils contain loess raw materials, the sand content can be defined as zero. The mean weight diameter (MWD) and mean geometric diameter (GMD) (mm) of each aggregate of each size were calculated as described by Wei et al. (2013) [37] as follows:

$$\text{MWD} = \sum X_i W_i \qquad (1)$$

where $W_i$ is the total sample weight fraction (%) remaining on the sieve, and $X_i$ is the mean diameter (mm), and

$$\text{GMD} = \exp\left\{\sum W_i \, lnX_i / \sum Wi\right\} \qquad (2)$$

where $W_i$ is the weight (g) of each aggregate.

### 2.3.3. Soil Organic Carbon (SOC)

The SOC concentration of the aggregate size fractions and the aggregate-associated C content and total organic C (OC) were determined using the $K_2Cr_2O_7$–$H_2SO_4$ digestion method [38]. SOC storage was calculated as a proportion of SOC (fixed depth method) following Zhang et al. (2015):

$$\text{Storage of } C_i = D_i \times \rho_i \times OC_i \times 10 \tag{3}$$

where $i$ is the soil layer ($i$ = 1, 2, and 3 represent the 10–20, 20–40, and 40–60-cm soil layers, –respectively); $OC_i$ is the SOC concentration (g C kg$^{-1}$) at each soil depth; $C_i$ is the SOC storage (Mg C ha$^{-1}$); $D_i$ is the thickness of the $i$th soil layer (m); and $\rho_i$ is the soil bulk density in the $i$th layer (g cm$^{-3}$) [39]. Ten is the soil layer thickness unit conversion factor, from centimeter (cm) to meter (m).

We also calculated the SOC stocks using the equivalent soil mass (ESM) method [40] to remove any unknown factors associated with the fixed depth (FD) method. The maximum weight of the fixed depth soil over the 4 treatments was determined as the equivalent soil mass. For the straw and humate treatments in lighter soil layers, an additional soil thickness ($T_{add}$) was required to reach the equivalent soil mass, which was calculated according to Zhang et al. (2015):

$$Tadd = (M_{soil;\ equivalent} - M_{soil;\ surface}) / BD_{subsurface} \times 10^4 \tag{4}$$

where $M_{soil;\ equivalent}$ is the equivalent soil mass (Mg ha$^{-1}$), $M_{soil;\ surface}$ is the sum of the soil mass from the surface layers (Mg ha$^{-1}$), and $BD_{subsurface}$ is the bulk density of the subsurface soil layers (Mg m$^{-3}$).

Then, the SOC stock of each aggregate size fraction based on ESM ($M_{equivalent}$, Mg ha$^{-1}$) was calculated as:

$$M_{equivalent} = M_{surface} + M_{Tadd} \tag{5}$$

where $M_{surface}$ is the SOC stock in the surface soil layer (i.e., if 0–20 cm is the surface layer, 20–40 cm is the subsurface layer, etc.), and $M_{Tadd}$ is the SOC stock in the additional subsurface soil layers (both Mg ha$^{-1}$). In our study, we used the soil weight of the blank treatment as the EMS, then $M_{equivalent}$ was calculated using Equation (4) with $D_i$ plus $T_{add}$. Since loess does not contain any gravel, the SOC stocks were given on a sand-free basis [41].

### 2.3.4. Soil Microbial Biomass Nitrogen and Carbon (MBN and MBC)

A mixed fresh soil sample (1 kg) from 5 cores of each plot with a soil profile of 0–20 cm was taken with an auger, immediately brought to the laboratory, and stored in a refrigerator at −4 °C. Then, the microbial masses of carbon and nitrogen (MBC and MBN) were determined by the gas extraction method [38] with a conversion factor of 0.38.

### 2.4. Crop Residue C Input

The crop residue C input (mainly from crop stubble, debris, and roots incorporated into the soil) was calculated according to the method reported by Malhi et al. (2011):

$$\text{BGR} = \text{a (GDW + AGR)} \tag{6}$$

$$\text{CR} = \text{AGR} + \text{BGR} \tag{7}$$

$$\text{CR-C} = \text{AGR-C} + \text{BGR-C} \tag{8}$$

where AGR is the aboveground residue, BGR is the belowground residue, GDW is the dry grain weight, CR is the cumulative crop residue, CR-C is the corresponding crop straw C returned to the soil, ARG-C is the amount of C in the aboveground residue, and BGR-C is the amount of C in the underground residue. The AGR and GDW were determined for the 2018 maize harvest with a water content of 14%. The BGR was estimated according to Formula (6). The C input was estimated by multiplying the C concentration by the C content

in the crop residue. The values of the constant *a* and the estimated C concentration were calculated for the maize residues based on the results of Gregorich et al. (2001), assumed to be 0.35. All crop residues in the soil were removed, except for the straw retention treatment (straw), so the C input of the crop residues from the other treatments was only due to the BGR-C.

### 2.5. Statistical Analyses

The data were analyzed using SPSS 19.0 software (IBM SPSS, Armonk, NY, USA) and compared to the results of the one-way ANOVA performed to assess the effects of the fully film-mulched ridge-furrow cultivation with a humate enhancement of the organic C sequestration. Linear regressions were used to calculate the aggregate size, stability (MWD and GMD), microbial biomass (MBC and MBN), and plant-derived carbon input (CR-C) into the soil carbon content and storage to determine their contribution to the soil carbon sequestration. LSD tests were conducted, with $p < 0.05$ considered significant. The diagrams were drawn using SigmaPlot 14.0 (Systat Software, Inc., Palo Alto, CA, USA).

## 3. Results

### 3.1. Soil Organic Carbon

#### 3.1.1. SOC Concentrations

At the end of the 3-year experiment, the SOC concentrations were higher ($p < 0.05$) in the humate and straw treatments and lower in the blank treatment compared to the controls. On average, higher soil carbon values were observed at the 0–60-cm soil depth in the humate and straw treatments, which were 18.0 and 11.7% higher compared to the control ($p < 0.05$). The SOC concentration showed a decreasing trend according to the mean soil depths of 0–20, 20–40, and 40–60 cm, but the straw and humate treatments were equally more effective than the others in increasing the SOC concentration, particularly at the depth of 0–40 cm (Table 2).

**Table 2.** Soil organic carbon (SOC) concentration, stock, and sequestration in 2018 after 3 years of experimentation.

| Item | Treatments | 0–20 cm | 20–40 cm | 40–60 cm | 0–60 cm |
|---|---|---|---|---|---|
| SOC (g kg$^{-1}$) | Blank | 6.86 c | 3.89 c | 3.04 b | 4.61 d |
| | Control | 6.90 c | 4.41 b | 3.34 ab | 4.91 c |
| | Humate | 7.3 b | 4.59 b | 3.59 a | 5.14 b |
| | Straw | 7.7 a | 5.21 a | 3.53 a | 5.44 a |
| SOC stock (Mg ha$^{-1}$) | Blank | 22.0 b | 10.8 b | 10.3 a | 43.1 b |
| | Control | 22.3 ab | 10.9 b | 10.4 a | 43.5 b |
| | Humate | 22.7 ab | 11.1 ab | 10.6 a | 44.4 ab |
| | Straw | 23.3 a | 11.6 a | 10.8 a | 45.4 a |
| SOC sequestration (Mg ha$^{-1}$ y$^{-1}$) | Blank | - | - | - | - |
| | Control | 0.3 (-) | 0.0 (-) | 0.1 (-) | 0.4 (-) |
| | Humate | 0.7 (0.4) | 0.3 (0.2) | 0.3 (0.2) | 1.3 (0.9) |
| | Straw | 1.3 (1.0) | 0. 8 (0.7) | 0.5 (0.4) | 2.3 (1.9) |

Note: SOC sequestration was increased in the SOC stocks in the straw, humate, and control treatments compared to the blank treatment and was calculated as the difference between the straw, humate, control, and blank. Blank: bare land; control: full film-mulched (FM) ridge-furrow tillage typical for maize production in the Loess Plateau using local standard chemical fertilizer doses of 150 kg ha$^{-1}$ N, 120 kg ha$^{-1}$ P$_2$O$_5$, and 75 kg ha$^{-1}$ K$_2$O; humate: FM with combined organic–inorganic fertilizer using active humic acid organic fertilizer (granules) of 1500 kg ha$^{-1}$ (organic matter/dry base matter ≥45.0–55%, free humic acid/dry base matter ≥20.0%); and straw: FM plus 7500 kg ha$^{-1}$ air-dried maize straw. Significant differences are indicated by lowercase letters ($p < 0.05$). Numbers in parentheses indicate the differences of the treatments in contrast to the control. Different letters (a, ab, b, c, d) in the same column indicate significantly different means according to LSD test at $p < 0.05$ level.

#### 3.1.2. SOC Stocks

The mean soil carbon stocks for the three tillage treatments were all higher compared to the blank treatment at the 0–20, 20–40, and 40–60-cm soil depths. Higher values without

significant differences were observed between the straw and humate treatments; both were significantly higher ($p < 0.05$) than the control and blank treatments, with increases of 4.4% for straw ($p < 0.05$) and 2.1% for humate ($p < 0.05$) compared to the control. Accordingly, sequestered SOC was higher in the 0–20-cm soil depth than in the 20–40 and 40–60-cm soil depths. High levels of SOC sequestration were observed with the straw and humate treatments at the 0–20, 20–40, and 40–60-cm soil layers and the 0–60-cm soil profile in contrast to the control (Table 2).

### 3.2. Soil Aggregates

#### 3.2.1. Aggregate Size Distribution

The depth distribution of the soil aggregates by dry sieving varied across the tillage methods and tillage depths (Figure 1). In the 0–60-cm layer, the humate and straw treatments increased the proportion of >2.0-mm soil aggregates almost equally significantly and reduced the proportion of soil aggregates <1.0 mm significantly ($p < 0.05$). No significant effects on the proportion of soil aggregates sized between 1 and 2 mm were observed. Although there were similar trends for each aggregate size, the tillage treatments affected the distributions observed at the three soil depths, which were more significant at 0–20 and 20–40 cm.

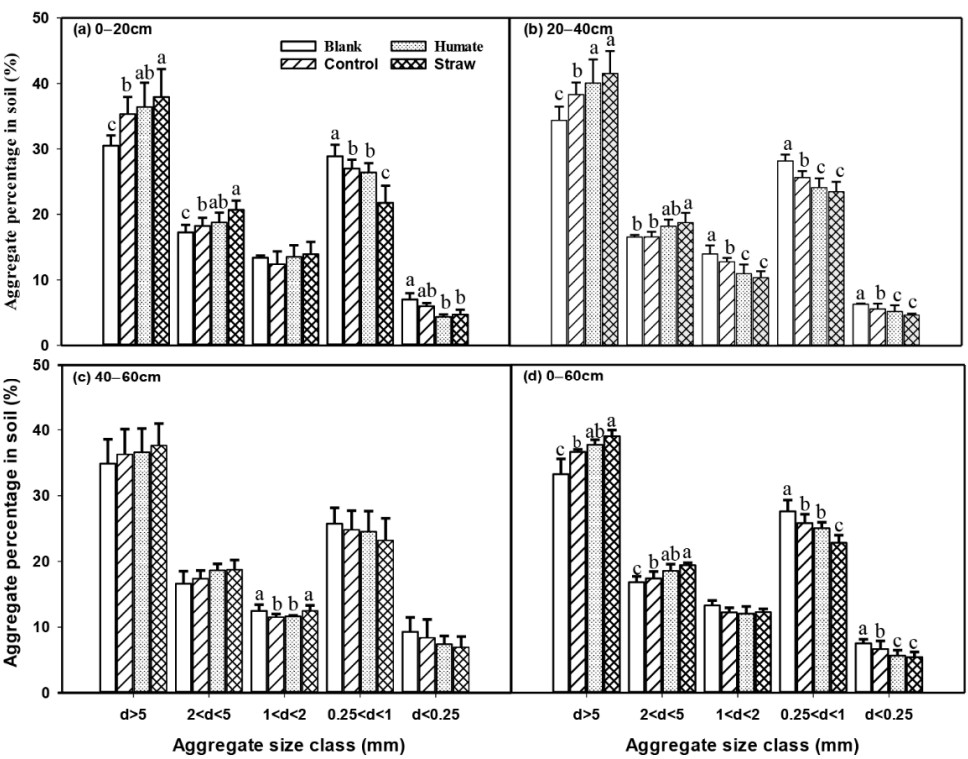

**Figure 1.** Percentage of soil aggregates (%) of different sizes in the soil layers 0–20 cm (**a**), 20–40 cm (**b**), 40–60 cm (**c**) and 0–60 cm (**d**) with dry sieving in 2018 after a 3-year trial. (**1**) Blank: bare ground, no agricultural treatment; (**2**) control: standard FM practice with standard fertilizer dose of 150 kg ha$^{-1}$ N, 120 kg ha$^{-1}$ P$_2$O$_5$, and 75 kg ha$^{-1}$ K$_2$O (serving as CK); (**3**) humate: FM plus humate, i.e., commercial humic organic fertilizer (granules) of 1500 kg ha$^{-1}$ (organic matter/dry basis matter ≥45.0–55.0%, free humic acid/dry basis matter ≥20.0%); and (**4**) straw: FM with straw retention, i.e., 7500 kg ha$^{-1}$ air-dried corn straw. Significant differences are indicated by lowercase letters (a, b, c) according to LSD's test at $p < 0.05$ level. Error bars represent the standard deviation (SD).

#### 3.2.2. Mean Weight Diameter (MWD) and Mean Geometric Diameter (GMD)

After 3 years of the trial, the MWD and GMD were strongly influenced by the additional organic treatments at 0–60 cm (Table 3). The MWD and GMD were increased at all soil depths ($p < 0.05$) compared to the blank treatment and were most increased in the humate and straw treatments. These improvements were most notable at 0–20 and 40–60 cm,

with the MWD and GMD increasing from 3.8 to 17.9% and 11.1 to 22.1%, respectively. The MWD and GMD values in the humate and straw treatments were similarly higher and not significantly different from those of the control, and all were significantly higher than those of the blank treatment ($p < 0.05$).

**Table 3.** The MWD and GMD of the soil aggregates by dry sieving in the different tillage systems at the end of the 3-year experiment in 2018.

| Treatments | MWD (mm) | | | | GWD (mm) | | | |
|---|---|---|---|---|---|---|---|---|
| | 0–20 cm | 20–40 cm | 40–60 cm | 0–60 cm Average | 0–20 cm | 20–40 cm | 40–60 cm | 0–60 cm Average |
| Blank | 2.6 ± 0.3 b | 2.8 ± 0.2 a | 2.6 ± 0.4 b | 2.7 ± 0.02 c | 1.8 ± 0.2 b | 2.0 ± 0.1 ab | 1.7 ± 0.4 b | 1.8 ± 0.1 c |
| Control | 2.8 ± 0.1 ab | 2.9 ± 0.2 a | 2.9 ± 0.2 ab | 2.9 ± 0.01 b | 2.0 ± 0.1 ab | 2.0 ± 0.2 ab | 2.0 ± 0.2 ab | 2.0 ± 0.1 bc |
| Humate | 2.8 ± 0.1 ab | 2.9 ± 0.3 a | 3.0 ± 0.2 ab | 2.9 ± 0.01 b | 2.0 ± 0.0 ab | 2.1 ± 0.2 ab | 2.1 ± 0.2 ab | 2.0 ± 0.0 b |
| Straw | 3.1 ± 0.2 a | 3.1 ± 0.3 a | 3.2 ± 0.0 a | 3.1 ± 0.01 a | 2.3 ± 0.2 a | 2.2 ± 0.3 a | 2.4 ± 0.1 a | 2.3 ± 0.1 a |

MWD, weight mean diameter, and GMD, geometric mean diameter. Different letters (a, ab, b, bc, c) in the same column indicate significantly different means according to LSD test at $p < 0.05$ level.

### 3.2.3. Aggregate-Associated Carbon

The aggregate-associated C content increased in the >1.0-mm soil aggregate fraction and decreased in the <0.25-mm fraction; however, there was no significant change in the fractions between 0.25 and <1 mm, independent of the tillage treatments in the 0–60-cm soil layer (Figure 2). Similar trends were observed at depths of 0–20 and 40–60 cm, but they differed from the effects observed at 20–40 cm, suggesting that organic improvement increases the aggregate-associated C content in soil fractions >1.0 mm but not in soil fractions <1.0 mm. In addition, significant increases in the SOC stocks of >0.25-mm soil aggregates were observed in both the straw and humate treatments, calculated using the EMS method (Table 4). All were significantly increased ($p < 0.05$) after the humate and straw treatments in contrast to the control and blank treatments.

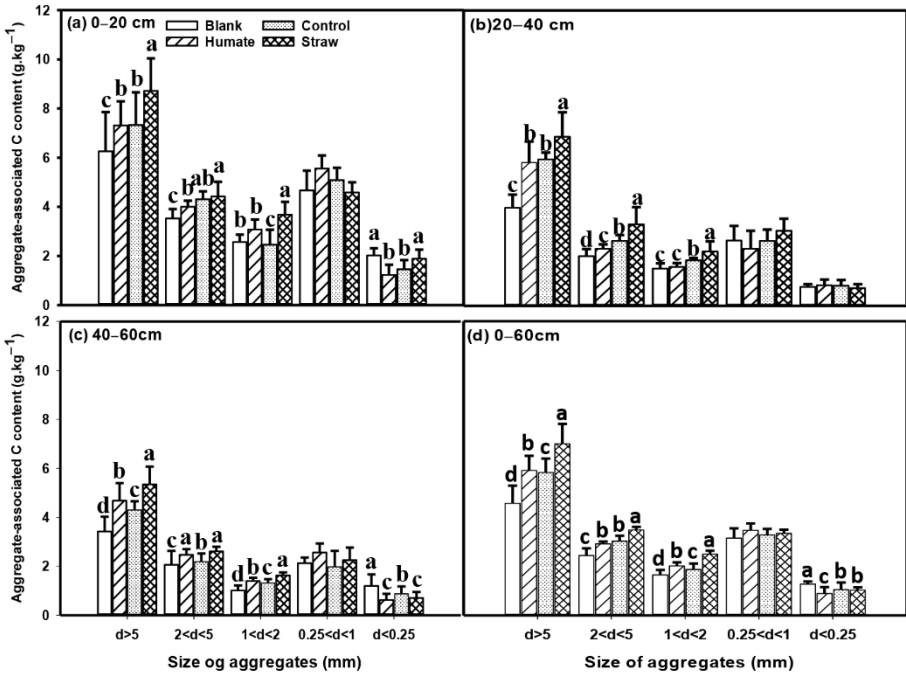

**Figure 2.** SOC concentrations in different soil layers 0–20 cm (**a**), 20–40 cm (**b**), 40–60 cm (**c**) and 0–60 cm (**d**) estimated using the fixed depth method in 2018 at the end of the 3-year experiment. Significant differences are indicated by lowercase letters (a, b, c, d) according to LSD test at $p < 0.05$ level. Error bars represent the standard deviation (SE).

**Table 4.** SOC stock in the 0–60-cm soil profile estimated by the equivalent soil mass (EMS) method at the end of the 3-year experiment in 2018.

| Tillage | SOC Stock (Mg ha$^{-1}$) | | | | | | Soil Mass (Mg ha$^{-1}$) | Tadd (cm) | SOC Stock in Additional Soil Layer (Mg ha$^{-1}$) | | | | | |
|---|---|---|---|---|---|---|---|---|---|---|---|---|---|---|
| | >5 mm | 2–5 mm | 1–2 mm | 0.25–1 mm | <0.25 mm | Total | | | >5 mm | 2–5 mm | 1–2 mm | 0.25–1 mm | <0.25 mm | Total |
| Straw | 9.4 ± 0.4 a | 9.6 ± 0.5 a | 9.6 ± 0.7 a | 8.8 ± 0.6 a | 11.3 ± 1.2 a | 48.7 ± 2.6 a | 8900.0 ± 511.6 c | 2.1 | 0.6 | 0.5 | 0.6 | 0.7 | 0.9 | 3.3 |
| Humate | 9.2 ± 0.2 a | 9.4 ± 0.4 a | 9.1 ± 0.3 b | 8.4 ± 0.7 b | 10.8 ± 1.0 ab | 46.7 ± 1.9 ab | 9066.7 ± 213.9 c | 1.6 | 0.6 | 0.6 | 0.4 | 0.4 | 0.2 | 2.3 |
| Control | 8.9 ± 0.8 c | 9.0 ± 1.2 | 8.7 ± 0.9 b | 8.0 ± 1.3 b | 10.5 ± 0.6 b | 45.1 ± 4.2 b | 9206. 7 ± 273.0 b | 1.0 | 0.6 | 0.4 | 0.2 | 0.3 | 0.1 | 1.6 |
| Blank | 8.1 ± 1.0 d | 8.5 ± 1.3 d | 8.2 ± 1.0 c | 7.8 ± 0.5 c | 10.1 ± 0.8 b | 43.1 ± 4.3 c | 9740.0 ± 156.2 a | - | - | - | - | - | - | - |

Note: Mean values ± SE. Tadd: additional soil thickness (m) required to attain the equivalent soil mass (ESM), calculated as $Tadd = (M_{soil; equivalent} - M_{soil; surface})/BD_{subsurface} \times 10^4$, where $M_{soil; equivalent}$ is the equivalent soil mass (mg ha$^{-1}$), $M_{soil; surface}$ is the sum of the soil mass in the surface layers (mg ha$^{-1}$), and BD $_{subsurface}$ is the bulk density of the subsurface soil layers (mg m$^{-3}$). Then, the SOC stocks of each aggregate size fraction based on ESM ($M_{equivalent}$, mg ha$^{-1}$) were calculated as: $M_{equivalent} = M_{surface} + M_{Tadd}$, where $M_{surface}$ is the SOC stocks in the surface soil layers (i.e., if 0–20 cm is the surface layer, 20–40 cm is the subsurface layer, and so on; mg ha$^{-1}$), and $M_{Tadd}$ is the SOC stocks in the additional subsurface soil layers (Mg ha$^{-1}$). Different letters (a, ab, b, c, d) in the same column indicate significantly different means according to LSD test at $p < 0.05$ level.

### 3.3. Soil Microbial Biomass Carbon and Nitrogen

The highest MBC and MBN values were observed with the straw treatment and were significantly higher compared to the other three tillage treatments (Figure 3). The effects of the tillage treatments on the MBC and MBN showed consistent trends of straw > humate > control > blank. Only the values for the MBC and MBN under the straw and humate treatments were significantly higher than those under the control, suggesting that the MBC and MBN increased in response to the straw and humate treatments.

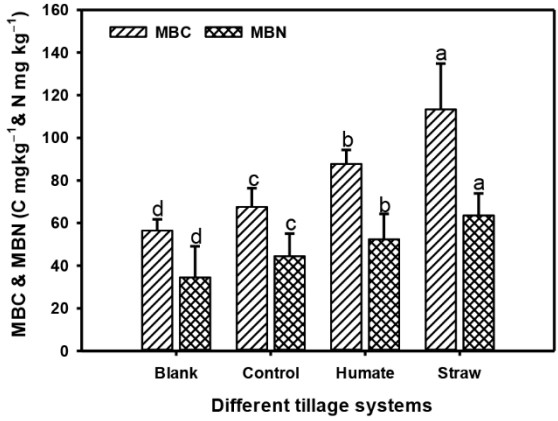

**Figure 3.** Influence of the humate and straw treatments on the MBC and MBN of the 0–20-cm soil layer in 2018 at the end of the 3-year trial. Significant differences are indicated by lowercase letters (a, b, c, d) according to LSD test at $p < 0.05$ level. Error bars represent the standard deviation (SE).

### 3.4. Contribution of the Aggregate-Associated C, MWD, GMD, MBC, MBN, CR-C, and Total Biomass Yield to Soil Carbon Storage

The increased soil carbon accumulation rate in the >0.25-mm soil aggregate fractions (Figure S1a–d) contributed most significantly ($R^2 = 0.88–0.97$ **, $p < 0.01$) to the total soil carbon concentration, which accounted for approximately 57% of the increase in the soil carbon storage. These then greatly increased the aggregate-associated C storage and significantly promoted the total soil carbon stores, with aggregate-associated C-storage > 1.0 mm being significantly ($R^2 = 0.79$ * $- 0.91$ ** and $p < 0.05$ and 0.01; Figure S2) correlated with the total soil carbon stocks, accounting for 22–58%. This indicates that straw and humate additives strongly supported the formation of macroaggregates, which then emitted more C, thereby significantly increasing the soil C storage. In addition, CR-C contributed significantly to the total soil carbon stocks ($R^2 = 0.88$ **, $p < 0.01$), as did the MWD, GMD, and MBN ($R^2 = 0.68–0.77$ *, $p < 0.05$), indicating that the crop C input, soil aggregate stability, and microbial biomass nitrogen have a strong impact on the soil carbon stocks (Figure 4a–e). In addition, the soil carbon stocks also correlated strongly with the total biomass yield

($y = 2.21 \times 85.50$, $R^2 = 0.91$ **, $p < 0.01$), suggesting that the soil carbon was unsaturated (Figure 4f).

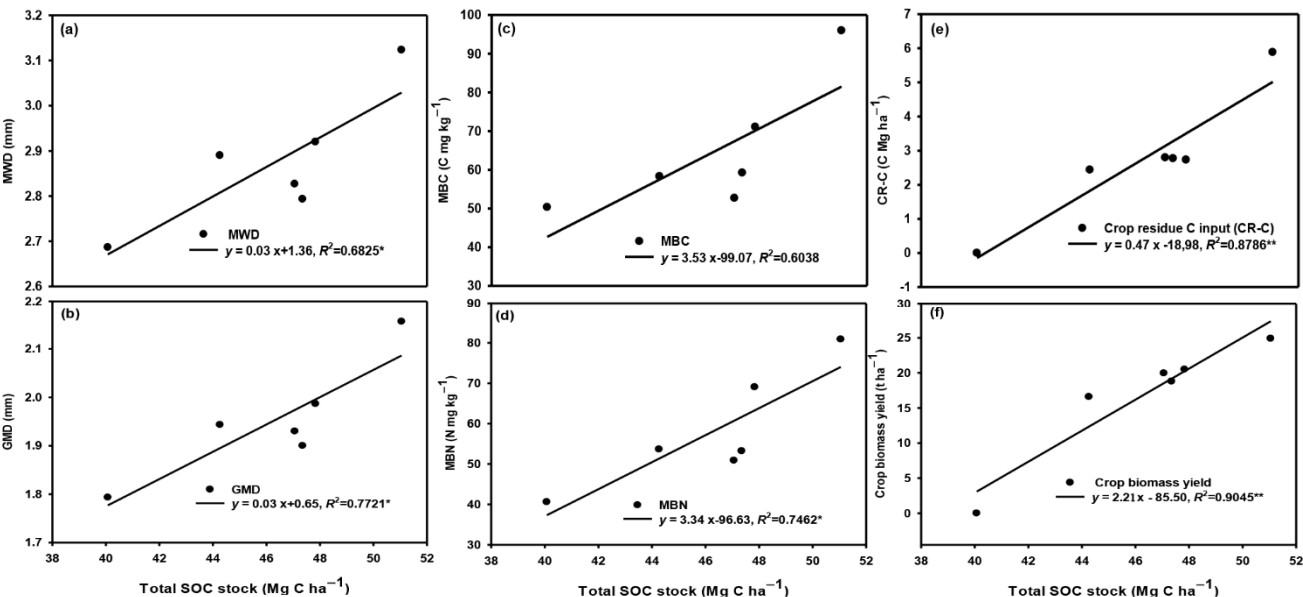

**Figure 4.** Correlation between the total SOC inventory and (**a**) the MWD, (**b**) GMD, (**c**) MBC, (**d**) MBN, (**e**) C use of crop residues (CR-C), and (**f**) total biomass yield. Linear regressions were performed using SPSS 19.0 software (IBM SPSS, Armonk, NY, USA). The graphs were generated using SigmaPlot 14.0 (Systat Software, Inc. Palo Alto, CA, USA). Significant differences were assessed by LSD testing at $p < 0.05$, * and ** were the significance at 0.05 and 0.01.

## 4. Discussion

### 4.1. Humate and Straw Affects Carbon Concentration in the Soil

In this study, as shown in Table 2, the humate and straw additions combined with FM tillage-altered soil carbon dynamics the most in the short term. This was comparable to the results in several previous studies [10,21,31,42]. One possible mechanism could be that humate and straw are themselves carbon-rich organic substances that are incorporated into the soil and increase the organic C content in the soil [6,14,41,43]. Second, these combined tillage patterns improved the soil temperature and moisture content, as shown in Figures S3 and S4 and reported in [44,45], which increased the plant growth and resulted in more residual carbon (litter and roots) being returned to the soil [21,46,47], which then compensated for the loss of soil carbon-related mineralization and uptake by plant growth. This would increase the soil carbon content in the soil [2,32]. Finally, the increased growth rates of the soil microbes (as shown in Figure S3) due to the additional energy from organic carbon and the modified hydrothermal conditions (Figures S4 and S5), dominated the decomposition of straw and humate, releasing more carbon and maintaining stable levels of microbial biomass carbon [9,22,42,48]. In the present study, the addition of straw and humate was more effective at increasing the soil carbon sequestration than the use of NPK fertilizer (control) alone or the blank treatment, with most changes in the concentrations and stocks occurring in the range from 0 to 20 cm (Table 2). The humate and straw treatments were the same at increasing the SOC stocks, suggesting that the two can be used as alternatives, depending on the available material.

### 4.2. Aggregate-Associated C and Physical Protection Are Major Contributors to C Sequestration

The formation, transformation, and depth distribution of soil aggregates are of great importance for SOC sequestration [17,49]. In our study, we found that the combinations of humate and straw additives with FM tillage significantly increased the macroaggregate fraction >2 mm and the associated C and increased the aggregate stability in the 0–40-cm

soil layer (Figure 1 and Tables 2 and 3), which partially updated the results of several previous studies [11,19,50,51] on fluvo-aquic soils in a similar area of Northwest China and on sandy loam soils in India. This could be due to the increase in the soil carbon content after the addition of humate and straw and the subsequent increased formation of organic cementitious materials (e.g., increased formation of macroaggregates and aggregate stability [25,52]), which have strong protective effects on the increased carbon sequestration in soil [12,49,52]. As shown in Figures 1 and 2, the C concentrations and stores associated with aggregates >1.0 mm correlated significantly with the soil carbon inventory, suggesting that the carbon in macroaggregates, promoted by the addition of straw and humate, significantly increased the soil carbon sequestration.

The increased physical protection of the aggregates and the increased decomposition of added straw and humate led to a higher release of organic C into the soil, which promoted the soil carbon sequestration. In this study, the treatments with straw and humate significantly increased the MWD in the 0–60-cm layer (Table 3), indicating an improved aggregate stability [11,53,54]. Increased aggregate stability strongly protects the soil carbon from decomposition [55,56], which increases the soil carbon sequestration [5,12,57,58]. Our data showed a significant correlation between the increased percentage of macroaggregates, aggregate stability, and carbon storage ($R^2 = 0.78$ * $-0.91$ **, $p < 0.05$ and 0.01) and total soil carbon stocks (Figure S2), indicating that aggregate stability and its associated C play an important role in increasing the soil carbon sequestration and maintaining the soil carbon balance in dry fluvo-aquic soils through straw and humate improvements.

### 4.3. Regulation of Microbial Diversity and Function Related to C Sequestration

Microbial processes regulate the carbon cycle in soils [45,59]. Previous studies have shown that plastic film mulching increases the activity of microorganisms by improving the soil hydrothermal and nutritional conditions [9,44,60]. This creates larger amounts of biomass and microbial residues that result from microbial regeneration and metabolism and contribute to the increased soil carbon content [15,61]. In our study, we found that the soil water content of the 0–100-cm soil profile and the temperature of the top 5-cm soil layer under the humate and straw treatments were significantly and considerably higher than those of the control or blank treatment (Figures S4 and S5); this can help increase the microbial activity in the soil and thereby increase the carbon levels. We also examined the soil bacterial communities and diversity using 16S rDNA Illumina pyrosequencing and found that the diversity of the soil bacteria was reduced under the treatments with humate and straw, but the relative abundance of some functional soil bacteria, such as Proteobacteria, Firmicutes, Actinobacteria, Gemmatimonadetes, Bacteroidetes, and Acidobacteria, increased (Supplementary Tables S1 and S2 and Figure S3). Downregulating the soil microbial diversity could help regulate the microbial process to control the breakdown of organic matter and leave more MBC in the soil, which could help to improve the soil aggregation and stabilization [48,56], thereby increasing the SOC levels in the soil [3]. Meanwhile, due to their increasing richness, some functional bacteria can act as plant growth-promoting rhizobacteria to enhance vigorous crop growth, resulting in more residual C returning to the soil. These could ultimately contribute to soil carbon accumulation. In this study, we observed that the highest MBC and MBN values increased after the straw and humate treatments (Figure 3). We also observed that the MBC and MBN were positively correlated with the soil carbon stocks (Figure 4), but the most notable associations were with the soil carbon and MBN ($R^2 = 0.77$ *, $p < 0.05$). These data indicate that increasing the MBC and MBN contributed to increased degradable organic carbon and, consequently, higher soil carbon values [8,48,62].

### 5. Conclusions

The combination of straw and humate with full film-mulched ridge-furrow cultivation significantly increased the SOC concentration and stocks, which have a major impact on the SOC dynamics in the dry fluvo-aquic soils of Northwest China. One possible mechanism is

that they are prone to forming a higher percentage of >2-mm soil macroaggregates, which trap the predominantly added organic carbon in >1-mm microaggregates, which, in turn, increases the aggregate-associated C content. At the same time, the soil carbon protection was further enhanced by increasing the stability of the soil aggregates, which also increased the soil carbon sequestration. Although the straw retention treatment sequestered relatively more C than the humate treatment, the increasing effect on the soil carbon concentrations and stocks was not significantly different in the three soil depths and the 0–60-cm soil profile, suggesting that the two can be used as alternatives. Therefore, the straw and humate additives offer the best options for long-term agricultural management practice to sequester soil carbon in the arid fluvo-aquic soils of Northwest China and are potential alternatives, depending on the straw and humate availability.

**Supplementary Materials:** The following supporting information can be downloaded at https://www.mdpi.com/article/10.3390/agronomy12061398/s1, Figure S1. Relationship between SOC and organic carbon content of different soil aggregate sizes at 0−60 cm soil layer. Figure S2. Correlation between total SOC stock and the different fractions of soil organic C in 0−60 cm soil layer. Figure S3. The relative abundance of the top 15 soil bacteria at phylum level under different treatments determined by using High-throughput 16S rRNA sequencing in 2018 (by OE Biotech Co., Ltd., Shanghai, China). Figure S4. The average soil water content of 0−100 cm soil layer in the period 2016−2018. Figure S5. The average soil temperature of the top 5 cm soil layer in the period 2016−2018. Table S1. The Chao1, Shannon and Simpson indices of alpha diversity of soil bacteria as affected by the treatments determined using High-throughput 16S rDNA sequencing in 2018 (by OE Biotech Co., Ltd., Shanghai, China). Table S2. The relative abundance of the top 15 soil bacteria at the phylum level under different treatments determined by High-throughput 16S rDNA sequencing in 2018 (by OE Biotech Co., Ltd., Shanghai, China.

**Author Contributions:** Conceptualization, F.Y. and B.H.; methodology, F.Y.; software, F.Y.; validation, F.Y., B.H. and G.Z.; formal analysis, F.Y. and G.Z.; investigation, B.H. and G.Z.; resources, B.H.; data curation, F.Y.; writing—original draft preparation, F.Y. and B.H.; writing—review and editing, F.Y.; visualization, F.Y. and G.Z.; supervision, F.Y.; project administration, F.Y.; funding acquisition, F.Y. All authors have read and agreed to the published version of the manuscript.

**Funding:** This research was funded by the National Natural Science Foundation of People's Republic of China, grant number 31860131 and 31560137; the Gansu Science and Technology Project, People's Republic of China, grant number 18YF1NA095-2.

**Acknowledgments:** We thank the many collaborators who are not listed as coauthors but who were involved in maintaining the field trials and collecting the soil samples.

**Conflicts of Interest:** The authors declare no conflict of interest.

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
