# Peer review of "Humate Combined with Film-Mulched Ridge-Furrow Tillage Improved Carbon Sequestration in Arid Fluvo-Aquic Soil"

_agronomy, doi:10.3390/agronomy12061398_

Round 1
Reviewer 1 Report
My comments can be found in the manuscript

Reviewer 2 Report
In fact, the effects of the addition of organic matter at 1,500 kg ha (humate) and maize straw (7,500 kg/ha) were investigated. Nota bene, the information on the kind of straw is given only in the note to the table 2 (as maize straw) and Figure 1 (as corn straw). It should be unified and given in the Methods chapter.
More information on humate should be provided. Especially that commercial humic products are usually applied in the form of a solution, the dosage of which is much lower, however stimulative for roots development. Minor corrections to the text are as follows:
49 correct the space bar
61 correct the “from the” for the “from the”
63-67 too long sentence should be divided into shorter sentences
92 the explanation to the Table 1 says “Lowercase letters represented significant differences between treatments “. However, table 21 presents only data for different layers of the same soil, and no different treatments.
105 correct the “stalks. 4 treatments” for the “stalks. The four treatments”
109 correct the “P2O5” for the “P2O5”
132 correct syntax error
143-146 Only SOC is measured, while SOM is the effect of converting SOC multiplying by the factor. This conversion does not make sense as the SOM can differ in C contents. I propose to remove the SOM (Omi) calculations.
147 The units in which the thickness of the layer (Di) is expressed are not specified. Judging from the formula, this value must be expressed in meters.
169-187 The whole paragraph „Crop residue C input” is not clear and requires careful checking and improving
176 there is something wrong with the Formula (7)
182 according to the numbers of the formulas, BGR was not estimated according to the Formula (6), but to the Formula (7)
186 correct the “thesoil” for the “the soil”
208 it seems that all data of SOC stock in the table should be check, because the SOC stock can be assessed, as SOC concentration given in 1 g/kg x 3. How was C Sequestration calculated?
210 correct the “I.e” for the “i.e.”
221 this sentence is trivial, because SOC stocks is calculated from SOC concentration
227 this sentence is also trivial, because SOC stocks is calculated from SOC concentration
228 it is obvious, that the sequestration SOC was higher at 0-20 cm soil layer than in deeper layers after addition of 1,500 kg/ha humate as well as 7,500 kg/ha maize straw.
260-263 this sentence is not clear
269 to avoid misunderstanding, in methods chapter the information should be provided on how aggregate-associated C-content was determined
275 correct the “mmbut” for the “mm but”
278 correct the “p <005)after” for the “p <005) after”
280 table 4 is hard to read. It would be better to divide it for two tables
293 at the Figure 2 the order of variants should be the same as in 0.Figure 1, that is: blank; control; humate; straw
307 it was 0-30 cm layer or 0-20 cm layer?
316 correct the “Cstorage” for the “C storage”
337 change “Humate affects” for “Humate and straw addition affects”
348 correct the “Fig s3)due” for the “Fig s3) due”
357 to be consequently, correct the “Aggregate-occupied C” for the “Aggregate-associated C”
